# Exploring the Putative Involvement of MALAT1 in Mediating the Beneficial Effect of Exendin-4 on Oleic Acid-Induced Lipid Accumulation in HepG2 Cells

**DOI:** 10.3390/biomedicines13020370

**Published:** 2025-02-05

**Authors:** Olfa Khalifa, Sama Ayoub, Abdelilah Arredouani

**Affiliations:** 1Diabetes Research Center, Qatar Biomedical Research Institute, Hamad Bin Khalifa University, Qatar Foundation Doha, Doha P.O. Box 34110, Qatar; okhalifa@hbku.edu.qa; 2Weill Cornell Medicine Qatar, Qatar Foundation, Doha P.O. Box 24144, Qatar; sna4001@qatar-med.cornell.edu; 3College of Health and Life Sciences, Hamad Bin Khalifa University, Qatar Foundation, Doha P.O. Box 34110, Qatar

**Keywords:** NAFLD, steatosis, exendin-4, GLP-1RA, GLP-1, MALAT1

## Abstract

**Background/Objectives**: The reduction of oleic acid (OA)-induced steatosis in HepG2 cells observed upon treatment with the glucagon-like peptide-1 receptor agonist (GLP-1RA) Exendin-4 (Ex-4) is associated with the modulation of the expression of several microRNAs, long non-coding RNAs (lncRNAs), and mRNAs. Notably, MALAT1, an lncRNA, shows significant downregulation in the presence of Ex-4 as compared to OA alone. In this study, we aimed to explore the role of MALAT1 in the positive impact of Ex-4 on OA-induced lipid accumulation in HepG2 cells. **Methods**: Steatosis in HepG2 cells was induced by treating them with 400 µM OA. The effect of Ex-4 on steatosis was examined by treating the steatotic cells with 200 nM of EX-4 for 3 h. MALAT1 was silenced with siRNA, while gene expression was quantified using qRT-PCR. **Results**: In the presence of Ex-4, the silencing of MALAT1 did not exert any discernible influence on de novo lipogenesis genes such as PPARγ and SREBP1. However, MALAT1 silencing significantly affected, to varying degrees, the expression levels of several lipid metabolism genes such as FAS, ACADL, CPT1A, and MTTP. **Conclusions**: Further investigations are warranted to fully decipher the role of the Ex-4-MALAT1 in the positive impact of GLP-1RAs on steatosis.

## 1. Introduction

Non-alcoholic fatty liver disease (NAFLD) is the most common chronic liver disease worldwide. It is driven mainly by the globally skyrocketing obesity epidemic and insulin resistance (IR) [1]. Approximately 25% of adults worldwide have NAFLD [2]. Regionally, the Middle East exhibits a higher prevalence, reaching 31.8%, while Africa demonstrates a comparatively lower prevalence at 13.5% [3]. Type 2 diabetes (T2D) and NAFLD commonly coexist, with a prevalence rate of 59.67% for NAFLD among individuals with T2D. NAFLD is a consequence of hepatic steatosis or the accumulation of triglycerides (TGs) as intrahepatic fat in more than 5% of the hepatocytes. It develops without additional factors such as alcohol intake. NAFLD encompasses a range of liver conditions, starting from harmless and reversible steatosis to non-alcoholic steatohepatitis (NASH), which may advance to liver cirrhosis, fibrosis, and potentially hepatocellular carcinoma. The mechanisms underlying non-alcoholic steatosis are poorly understood, and minimal knowledge is available on the pathways involved in progressive hepatocellular damage caused by the accumulation of lipids.

As of the present moment, the absence of an approved therapeutic modality for NAFLD persists. The currently available interventions that have demonstrated efficacy in improving hepatic function, reducing the severity of NAFLD, and enhancing glycemic control and vascular function rely primarily on implementing dietary modifications and engaging in exercise-induced weight reduction [4,5,6,7,8]. However, achieving and maintaining weight loss is notoriously challenging for most patients. Therefore, there is an urgent and unmet need for effective pharmacotherapy. Evidence from recent investigations has suggested the potential therapeutic candidacy of glucagon-like peptide-1 receptor agonists (GLP-1RAs) for NAFLD both in humans [9,10,11,12,13] and in animal models [14,15]. The GLP-1RAs are widely recognized for their ability to induce weight loss [16,17,18] and have already been approved for the management of T2D [19]. Nonetheless, the precise mechanisms underlying the observed protective effect of GLP-1R agonists on steatosis remain to be unraveled. Both indirect mechanisms, involving weight loss, and direct mechanisms, involving the activation of hepatic GLP-1R and the downstream signaling pathways, have been proposed to elucidate the beneficial impact of GLP-1RAs on hepatic steatosis [20].

Recently, we proposed that the direct activation of GLP-1R by Ex-4 mitigates steatosis induced by oleic acid (OA) in HepG2 cells. This effect is achieved by reducing the expression of the Fatty Acid Binding Protein 1 (FABP1) and the Forkhead box protein A1 (FOXA1) through the modulation of the Wnt/β-catenin signaling pathway [21]. Previous studies have suggested the involvement of the Wnt/β-catenin signaling pathway in the anti-steatotic effect of Ex-4 on HepG2 cells by inhibiting lipogenesis [22]. Additionally, β-catenin has been implicated in mediating the beneficial effect of exenatide, another GLP-1RA, on ameliorating hepatic steatosis induced by a high-fructose diet in rats [23]. The NAD-dependent deacetylase sirtuin-1 (SIRT1) has also been proposed to play a role in the effect of exenatide on improving hepatic steatosis in both in vitro and animal models [24,25]. Furthermore, recent work by Yu et al. suggests that Ex-4 attenuates hepatic steatosis in HepG2 cells by promoting the autophagy–lysosomal pathway [26].

To further explore the molecular mechanisms underpinning the beneficial effects of GLP-1RAs on steatosis, we recently investigated the modulation of the expression of microRNAs (miRNAs) and long non-coding RNAs (lncRNAs) upon the treatment of OA-induced steatotic HepG2 cells with Ex-4. Our findings showed significant reductions in OA-induced steatosis in response to Ex-4, accompanied by alterations in the expression of multiple miRNAs [27] and lncRNAs [28]. One notable lncRNA affected by Ex-4 treatment of OA-induced steatotic cells is metastasis-associated lung adenocarcinoma transcript 1 (MALAT1). MALAT1 is a highly conserved lncRNA implicated in various diseases, including cancer [29]. It has also garnered attention for its involvement in the pathogenesis of diabetes-related microvascular disease and diabetic retinopathy. Furthermore, MALAT1 has been shown to stimulate the production of inflammatory cytokines in endothelial cells treated with high glucose [30]. However, to date, no previous studies have investigated the role of MALAT1 in reducing hepatic steatosis in response to GLP-1RAs.

Recent research has highlighted the role of several lncRNAs in the development of NAFLD. For example, Ye et al. demonstrated that silencing the lncRNA NEAT1 could mitigate NAFLD progression by modulating the miR-129-5p/SOCS2 signaling pathway [31]. Similarly, Wang et al. found that the lncRNA H19 promoted hepatic lipogenesis in NAFLD by enhancing both the mTORC1 signaling pathway and the MLXIPL transcriptional network [32]. Recent studies have begun to explore the role of MALAT1 in lipid metabolism, particularly within the liver, where it may influence the development and progression of metabolic diseases such as NAFLD [33]. MALAT1 has been shown to modulate key regulatory pathways involved in lipid homeostasis, including the regulation of lipogenesis and lipid oxidation [34,35], making it a potential player in the pathogenesis of hepatic steatosis and related metabolic disorders. Furthermore, MALAT1 expression was found to be elevated in the liver tissues of NAFLD patients [36]. Additionally, FFA-induced lipid accumulation in hepatocytes can be largely reversed by the knockdown of MALAT1 [33].

While the in vitro results of the present study provide a promising foundation, the path to clinical application involves addressing the complexities of in vivo systems and ensuring that the applied treatments are both safe and efficacious in the context of the whole organism. Translating the in vitro findings to in vivo models or clinical settings presents both opportunities and challenges. This study’s results could potentially lead to the development of MALAT1-targeted therapies in combination with GLP-1RAs for enhanced NAFLD treatment, and MALAT1 levels might serve as a biomarker for disease progression or treatment response. However, the complex liver microenvironment, interactions with other organs, and potential long-term effects not observable in cell culture necessitate further investigation in animal models. These studies would allow for the assessment of systemic effects, tissue-specific responses, and long-term consequences of MALAT1 modulation and Ex-4 treatment. Clinical translation would require carefully designed studies to determine the appropriate dosage, delivery methods, and safety profiles.

The primary aim of this investigation was to assess the contribution of MALAT1 in mitigating OA-induced hepatic steatosis in HepG2 cells upon treatment with Ex-4. This objective was pursued by scrutinizing the transcriptional activity of many genes implicated in de novo lipogenesis, fatty acid uptake, transport, and overall lipid metabolism subsequent to the suppression of MALAT1 expression.

## 2. Materials and Methods

### 2.1. HepG2 Culture and OA Preparation

The human hepatoma HepG2 cell line (HB-8065, ATCC) was obtained from ATCC (Manassas, VA, USA) and cultured in Dulbecco’s modified Eagle’s medium (DMEM) (31966047, Gibco, MA, USA) at 37 °C and 5% CO_2_ with 10% FBS (10500064, Gibco, MA, USA) and 1% penicillin/streptomycin. The OA solution was prepared as previously reported [37]. Briefly, OA powder (O-1008 Sigma-Aldrich, Hamburg, Germany) was dissolved at a final concentration of 12 mM in phosphate-buffered saline (PBS; 137 mM NaCl, 10 mM phosphate, 2.7 mM KCl, and pH 7.4) that contained 11% fatty-acid-free bovine serum albumin (BSA; MP Biomedicals, Santa Ana, CA, USA) by sonification at a frequency of 10 kHz with two 5 min pulses before shaking at 37 °C for 15 h using an OM10 Orbital Shaking Incubator (Ratek Instruments Pty, Ltd., Boronia, Australia). The OA solution was filtered using a 0.22 µm filter and stored at 4 °C before use. We used a fresh aliquot for each experiment. All the experiments were carried out with cells passaged no more than 25 times.

### 2.2. Induction of Steatosis and Treatment with Exendin-4

We used the same procedure as in our previous publications to create the steatosis cell model and treat it with Ex-4 [21,27,28]. In brief, we cultured HepG2 cells in 6-well plates at a density of 4 × 10^5^ cells/well until 70% confluency, then starved them for six hours in DMEM containing 1% fatty-acid-free FBS. Following this starvation, we incubated the cells for 16 h at 37 °C in DMEM containing 400 µM OA and 1% fatty-acid-free FBS, and then we quantified the steatosis. We used 1% fatty-acid-free FBS in all OA treatment experiments to ensure that OA was the only inducer in the medium and that OA did not react with components of FBS. Following steatosis induction, we washed the cells with PBS and incubated them for three hours in fresh 1% FBS DMEM containing 400 µM OA solution in the absence or presence of 200 nM of Ex-4 (E7144-1MG, Tocris, Minneapolis, MN, USA). The optimal concentrations of OA and Ex-4 we used were determined in our previous paper [37]. Briefly, the Ex-4 concentration was determined by a dose–response experiment. Different concentrations (100 to 600 nM) and times (1 h to overnight) were examined, and the best results were obtained through treatment with 200 nM for 3 h. Longer durations of Ex-4 treatment and higher concentrations did not improve the outcome, likely because Ex-4 is degraded in the culture media after 3 h. For each experiment, we used a fresh aliquot of Ex-4. The cell viability was assessed, and throughout each phase of the experiments, cells exhibited viability levels ranging from 80% to 90%.

### 2.3. Quantification of Steatosis

As described in our previous studies, the steatosis was quantified by TG measurement using a commercial fluorometric test kit (Abcam TG quantification assay kit, Waltham, MA, USA, ab65336) and a microplate reader to detect total TG levels (Infinite F200 Pro; Tecan, Männedorf, Switzerland) [21,27,28]. We also used imaging of lipid droplets labeled with BODIPY 493/503, which labels neutral lipids. In this study, we measured the TG levels in HepG2 with and without MALAT1 transfection, and we also quantified the mRNA expression of three perilipin proteins associated with the surface of lipid droplets.

### 2.4. Total RNA Isolation and Real-Time PCR

Total RNA was extracted using a Pure Link RNA Mini kit (12183025, Invitrogen, Carlsbad, CA, USA), according to the manufacturer’s instructions. RNA concentrations were assessed using a NanoDropTM spectrophotometer (Thermo Fisher Scientific, Waltham, MA, USA). The RNA samples were immediately frozen at −80 °C until use. To prepare cDNA, a High-Capacity cDNA Reverse Transcription kit (4368813, Applied Biosystems, Foster City, CA, USA) was used. Gene expression was quantified using qRT-PCR on a QuantStudio 6 Flex system (ThermoFisher, Waltham, MA, USA) with PowerUpTM SYBR Green Master Mix (A25780, Applied Biosystems, Woburn, MA, USA), and the relative expression was normalized to that of β-actin as an internal control. The comparative 2^−ΔΔCT^ method was used to calculate the relative expression. The expression levels of the following genes were quantified: sterol regulatory element binding transcription factor 1 (SREBP-1), peroxisome proliferator-activated receptor gamma (PPARγ), Diacylglycerol O-acyltransferase 1 (DGAT1), Diacylglycerol O-acyltransferase 2 (DGAT2), fatty acid synthase (FAS), Acyl-CoA Dehydrogenase Long Chain (ACADL), Carnitine Palmitoyltransferase 1A (CPT1A), Stearoyl-CoA desaturase 1 (SCD-1), Acetyl-CoA carboxylase alpha (ACC), nuclear receptor subfamily 1 group H member 2 (NR1H2), Microsomal Triglyceride Transfer Protein (MTTP), and metastasis-associated lung adenocarcinoma transcript 1 (MALAT1). The primers we utilized for the genes are listed in Table 1. We designed specific primers using Primer-BLAST (https://www.ncbi.nlm.nih.gov/tools/primer-blast/, accessed on 1 March 2024) that met the following criteria: (1) Primer pairs are distinct. They do not bind to other genome parts besides the intended gene or DNA fragment. (2) Primer pairs do not bind together (forming a primer dimer): self- or hetero-dimer. (3) The possibility of the primers forming a secondary structure, which could interfere with PCR amplification, is very low. (4). The Tm (temperature of mismatch) values of two primers are intended to be close. (5) Tm is much lower than TA (annealing temperature). Furthermore, our primers were examined using the IDT company’s (Newark, NJ, USA) Oligo Analyzer 3.1 program (https://eu.idtdna.com/page) accessed on 1 March 2024.

### 2.5. Cell Transfection and MALAT1 Knockdown

For siRNA-mediated MALAT1 gene silencing, we transfected HepG2 cells with 20 nM of MALAT1-specific siRNA or negative scrambled siRNA, according to the manufacturer’s instructions. HepG2 cells were transfected with MALAT1-specific siRNA or a negative scrambled siRNA using Dicer-Substrate Short Interfering RNAs (DsiRNAs), TriFECTa^®^ Kits (http://www.idtdna.com/calc/analyzer, accessed on 1 March 2024), and the Lipofectamine RNAiMAX transfection kit (13778-075; Invitrogen, Carlsbad, CA, USA). The DsiRNAs-TriFECTa^®^ kit contains three Dicer-substrate 27-mer RNA duplexes specific for a single target gene. A pool of the three duplexes was used to silence MALAT1. After transfection, cells were cultured under normal growth conditions (37 °C, 5% CO_2_) for 24 h without antibiotics. The silencing efficiency was checked by quantifying the expression of MALAT with qRT-PCR. The expression of the following genes was also quantified: SREBP-1, PPARγ, FAS, ACADL, CPT1A, NR1H2, MTTP, ACC, SCD1, DGAT1, and DGAT2. During the optimization of our experiment, we included an untreated control to verify that the scrambled siRNA control and the untreated control produced similar effects. We validated that the scrambled siRNA control did not exhibit any off-target effects.

Data were normalized to those for β-actin as an internal control. Gene relative expression was calculated using the comparative 2^−ΔΔCT^ method.

### 2.6. Western Blotting

HepG2 cells were treated with OA and Ex-4, followed by lysis in buffer A (50 mM HEPES, pH 7.2, 150 mM NaCl, 1 mM EDTA, 1 mM EGTA, 20 mM NaF, 2 mM Na3VO4, 10 mM β-glycerophosphate, 1% Triton X-100, 10% glycerol, 1 mM PMSF, 10 mM sodium butyrate, 1% aprotinin, 0.1% SDS, and 0.5% sodium deoxycholate), and the lysates were clarified by centrifugation. For the transfection of HepG2 cells with malat1 scramble and malat1 siRNA, proteins were extracted using RIPA buffer from the PARIS™ Kit (AM1921, Ambion^®^ PARIS™, Invitrogen, Carlsbad, CA, USA). A total of 20 µg of protein was loaded onto 10% Tris-Glycine Mini Gels (Novex, XP00100BOX, Thermo Fisher Scientific) and transferred to a 0.2 µm PVDF membrane (Bio-Rad, Hercules, CA, USA) using the Trans-Blot Turbo system (Bio-Rad, Hercules, CA, USA). Following the transfer, the membranes were incubated overnight at 4 °C with primary antibodies: anti-SCD1 (Cell Signaling, Danvers, MA, USA, #2438), anti-PPAR γ (Santa Cruz, CA, USA, sc-7196), anti-SREBP-1 (Santa Cruz, CA, USA, sc-365513), anti-FAS (Cell Signaling, Danvers, MA, USA, C20G5), and anti-β-actin (Cell Signaling, Danvers, MA, USA #4970). After the membranes were washed three times with PBS, they were incubated with appropriate horseradish peroxidase-conjugated secondary antibodies. Membrane development was achieved using SuperSignal™ West Femto Maximum Sensitivity Substrate (34094, Thermo Fisher Scientific), and chemiluminescence was detected with the Bio-Rad ChemiDOC XRS system (Bio-Rad, Hercules, CA, USA). The band intensity was normalized to that of β-actin for the total protein and that of Lamin-B1 for nuclear protein levels. Antibody dilutions were used as recommended by the manufacturer, unless specified otherwise.

### 2.7. Statistical Analysis

We performed the statistical analysis and the graphing with GraphPad Prism 9.0 software (GraphPad Prism v9, La Jolla, CA, USA). Data are presented as the mean ± SEM. We used unpaired one-way ANOVA analysis (ANOVA) to assess the significance of differences in mean values between experimental groups, and Tukey’s post hoc test was used to adjust multiple comparisons between experimental groups. When we silenced MALAT1, we used a two-way analysis of variance (ANOVA) to evaluate the significance of differences between the mean values of different experimental groups. To analyze the images of Western blots, we used ImageJ software (version 1.8.0, NIH, Bethesda, MD, USA). Unless otherwise specified, a *p*-value of <0.05 was considered significant.

## 3. Results

### 3.1. Study Design

HepG2 cells were seeded in 6-well plates and grown to approximately 70% confluence. The cells were then incubated in DMEM without antibiotics (penicillin/streptomycin) for 24 h. Following this, we transfected the cells with 20 nM of MALAT1-specific siRNA or a negative control of scrambled siRNA, according to the manufacturer’s protocol. Transfection was performed using Dicer-Substrate Short Interfering RNAs (siRNAs) along with Lipofectamine RNAiMAX transfection reagent. After transfection, the cells were starved for several hours in DMEM containing 1% fatty-acid-free FBS. To induce steatosis, the cells were treated for 16 h at 37 °C with DMEM containing 1% FBS and 400 µM OA. After this induction, the cells were incubated for an additional 3 h in fresh DMEM containing 1% FBS and 400 µM OA, with or without 200 nM Ex-4. Following the treatments, the total RNA was extracted from the cells for further analysis. The gene expression was quantified under different experimental conditions. The data were subsequently analyzed using various bioinformatics tools, as shown in Figure 1, which was created using BioRender.

### 3.2. Exendin-4 Reduces OA-Induced Lipid Accumulation in HepG2 Cells

After steatosis induction, the cells were washed and incubated in fresh DMEM containing 400 μM OA in the absence or presence of Ex-4 (E7144-0.1MG, Tocris, Minneapolis, MN, USA). To determine the optimal concentration of Ex-4, we treated the steatotic cells with increasing concentrations of Ex-4 from 0 to 1 mM and with different incubation periods (3, 6, 12, and 24 h). We then quantified the TG content as above. We used a fresh aliquot of Ex-4 for each experiment.

In agreement with [21,38], the treatment of HepG2 cells with 400 µM OA induced a significant accumulation of TGs in lipid droplets. The steatotic cells exhibited a significant decrease in lipid accumulation in response to 3 h of treatment with 200 nM of Ex-4, as evidenced by a reduction in their TG content [21] and a downregulation in their expression of perilipin genes PLIN2 and PLIN3, which encode for proteins that coat the lipid droplets [39] (Figure 2).

### 3.3. Oleic Acid Treatment Upregulates MALAT1 in HepG2 Cells

To explore the involvement of MALAT1 in lipid accumulation, we treated HepG2 cells with increasing concentrations of OA (200–500 μM) and quantified the expression of MALAT1 using qPCR. As depicted in Figure 3A, OA significantly and dose-dependently elevated the expression of MALAT1 (*p* < 0.0001).

### 3.4. MALAT1 Knockdown Reverses OA-Induced Lipid Accumulation

As illustrated in Figure 3B, we achieved about 80% (*p* < 0.0001) MALAT1 silencing at the mRNA level. Interestingly, the silencing of MALAT1 in steatotic HepG2 cells significantly decreased the OA-induced lipid accumulation, as shown by the downregulation of PLIN genes (Figure 3C–F). MALAT1 mRNA expression was quantified in untreated HepG2 cells, steatosis-induced cells, OA+Ex-4-treated cells, and Ex-4-alone-treated cells. OA treatment significantly increased MALAT1 expression in HepG2 cells (*p* = 0.04). Conversely, treatment with OA+Ex-4 and Ex-4 alone significantly decreased MALAT1 expression in HepG2 cells (*p* = 0.026 and *p* = 0.03, respectively) (Figure 3G).

### 3.5. Exendin-4 Inhibits the Effect of OA on De Novo Lipogenesis Gene Expression in HepG2 Cells When MALAT1 Is Silenced

To further investigate the role of MALAT1 in the protective effect of Ex-4 on steatosis, we quantified the expression of the SREBP-1 and PPARγ genes, known for their role in de novo lipogenesis. In accordance with the data presented in Figure 4, the treatment with Ex-4 markedly impeded the elevation in SREBP-1 and PPARγ gene expression induced by OA. Notably, the silencing of MALAT1 did not exert any discernible influence on the impact of Ex-4.

### 3.6. Effect of MALAT1 Silencing on the Expression of Lipogenesis, Fatty Acid Uptake, and Transport Genes in HepG2 Cells in the Presence of Ex-4

We examined the expression of several lipogenesis genes, including DGAT1, DGAT2, FAS, ACADL, CPT1A, SCD1, NR1H2, MTTP, SLC2, and ACC, in response to OA and Ex-4 before and after silencing MALAT1. Compared to untreated HepG2 cells, steatotic cells showed a significant upregulation of the lipogenesis genes DGAT1, DGAT2, FAS, CPT1A, SCD1, ACC, NR1H2, and MTTP, while ACADL was downregulated and SCL2 expression was unaffected. Interestingly, when compared to OA alone, the presence of Ex-4 significantly reversed the upregulation of DGAT1, DGAT2, SCD1, ACC, and NR1H2 (Figure 5), while the expression of FAS, ACADL, CAP1A, MTTP, and SCL2 remained unaffected (Figure 5).

We performed additional investigations to evaluate the impact of MALAT1 on the abovementioned modulation of gene expression induced by OA and Ex-4. The silencing of MALAT1 resulted in diverse outcomes concerning the effects of OA and Ex-4 on gene expression. Specifically, the previously observed upregulation of DGTA1, DGTA2, ACC, and MTTP expression in response to OA treatment lost statistical significance following MALAT1 knockdown (Figure 5A,B,G,J). Similarly, the significant downregulation of DGTA2 and SCD1 induced by Ex-4 treatment was no longer evident upon MALAT1 silencing (Figure 5A,F). In contrast, MALAT1 knockdown rendered the Ex-4-mediated insignificant downregulation of FAS1 and CPT1A statistically significant (Figure 5C,E).

### 3.7. Modulation of Lipogenesis and Fatty Acid Protein by MALAT1 Silencing in HepG2 Cells Treated with Ex-4

To better understand the potential role of MALAT1 silencing as a molecular determinant through which Ex-4 mediates its beneficial effect on steatosis, we quantified SREBP1, PPRAδ, FAS, SCD1, ACC, and MTTP at the protein level before and after silencing. We could not detect ACADL and MTTP with the antibody we used, despite using up to 60 mg of protein and 1/200 antibody dilution (the company recommends 1/1000 dilution). The impact of Ex-4 on the PPARγ and SREBP1 protein levels contrasts with its impact on the mRNA, suggesting a posttranslational regulation that implicates the MALAT1 pathway. Compared to the scrambled siRNA transfection (Figure 6A,B), OA significantly decreased PPARγ and SREBP1 protein expression, relative to that in untreated cells, following MALAT1 knockdown (*p* = 0.033). However, the effect of Ex-4 on FAS and SCD1 protein expression, relative to that in OA-treated cells, was significantly decreased, with *p* = 0.001 and *p* = 0.002, respectively, between scrambled transfection and by Malat-1 knockdown (Figure 6C,D). Interestingly, Ex-4 significantly reduced the expression of FAS and SCD1 only after scrambled transfection knockdown (Figure 6C,D).

## 4. Discussion

The present study is the first to examine the role of the lncRNA MALAT1 in the protective effect of the GLP-1RA Ex-4 on steatosis induced by OA in HepG2 cells. NAFLD has become a significant global public health burden [40], with no approved pharmacological interventions for its management. Although GLP-1RAs have been suggested as a potential therapy for NAFLD, their mechanisms of action remain to be unraveled. Numerous studies have shed light on the role of lncRNAs as pivotal regulators in various physiological and pathological processes, including lipid metabolism and the development of NAFLD [41,42,43,44].

Consistent with our previous findings [28,45], the present study confirms that Ex-4 treatment significantly reduces OA-induced lipid accumulation in HepG2 cells. Additionally, our results reveal an upregulation of MALAT1 expression in steatotic cells and a significant decrease in lipid accumulation upon MALAT1 silencing, implying the involvement of this lncRNA in the lipid accumulation process. The present findings corroborate the observations documented by Yan et al. [35], wherein they reported an upregulation of MALAT1 expression in HepG2 cells following exposure to palmitate, as well as in the livers of ob/ob mice. Notably, their study demonstrated that suppressing the expression of MALAT1 markedly mitigated palmitate-induced lipid accumulation in HepG2 cells. Moreover, the knockout of MALAT1 in ob/ob mice resulted in a substantial reduction in liver lipids and concomitant enhancement of insulin sensitivity. Conversely, the overexpression of MALAT1 led to lipid accumulation, thereby implicating MALAT1 in the development of hepatic steatosis and insulin resistance [35]. Interestingly, an upregulation of MALAT1 was also observed in patients with NAFLD and was suggested to contribute to the progression of liver fibrosis [46].

Investigations examining the role of MALAT1 in NAFLD are scarce. For instance, MALAT1 was linked to liver inflammation through activation of the NLRP3 inflammatory complex in patients with chronic hepatitis B and NAFLD [47]. A role of MALAT1 in liver fibrosis via the regulation of the chemokine CXCL5 was also reported in a study investigating fibrosis in NASH patients [48]. Together, these studies imply that MALAT1 may play various roles in NAFLD development, especially as there is evidence that MALAT1 can modulate the lipid synthesis process [49]. Consistent with our previous findings [21], the current study demonstrated a notable increase in the expression of PPARγ and SREBP1 upon treatment with OA. Notably, the administration of Ex-4 effectively mitigated this upregulation. However, our investigation did not reveal any significant influence of MALAT1 silencing on the impact of OA and Ex-4 on the expression of PPARγ and SREBP1. These findings diverge from those reported by Xiang et al. [33], who demonstrated that knockdown of MALAT1 led to the downregulation of PPARγ and SREBP1 in HepG2 cells. We do not have an explanation for this discrepancy, but we note that while in our study, we used 400 µM of OA for 16 h to induce steatosis, Xiang and colleagues used a 1 mM mixture of oleate and palmitate (2:1 ratio) for 24 h.

Consistent with our prior investigation [21], our current findings demonstrate that the treatment of HepG2 cells with OA significantly upregulated the lipid-metabolism-associated genes DGTA1, DGTA2, FAS, CPT1A, SCD1, ACC, and NR1H2. Conversely, the ACADL and SCL2 genes were downregulated. Notably, introducing Ex-4 treatment effectively mitigated the effects of OA on DGTA1, DGTA2, SCD1, ACC, and NR1H2 gene expression. In essence, hepatic lipid accumulation can arise through heightened absorption of circulating free fatty acids (FFAs), elevated hepatic de novo lipogenesis, attenuated hepatic β-oxidation, and reduced hepatic lipid export via very-low-density lipoprotein (VLDL) secretion [50]. The silencing of MALAT1 yielded divergent outcomes regarding the effects of OA and Ex-4 on gene expression. Hence, the previously observed significant upregulation of DGTA1, DGTA2, ACC, and MTTP expression in response to OA treatment lost statistical significance following MALAT1 knockdown. Furthermore, the significant downregulation of DGTA2 and SCD1 induced by Ex-4 treatment was no longer evident upon MALAT1 silencing. Conversely, MALAT1 knockdown exacerbated the Ex-4-induced DGTA1 downregulation and rendered the Ex-4-mediated downregulation of FAS1 and CPT1A statistically significant when compared to OA alone. DGTA1 is a key enzyme for triglyceride synthesis, and increased levels of DGAT1 mRNA have been reported in human livers with NAFLD [51]. Moreover, global and liver-specific inactivation of DGAT1 in mice afforded protection from steatosis due to a high-fat diet [51]. The effect of MALAT1 silencing on DGAT1 expression in the presence of Ex-4 suggests that Ex-4 reduces OA-induced steatosis by a mechanism that implicates the deactivation of DGAT1, and, thus, TG synthesis, via the modulation of MALAT1.

The significant reduction in FAS1 upon MALAT1 silencing is an indication that Ex-4 might reduce de novo lipogenesis by downregulating MALAT1. NR1H2, also called liver X receptor beta (LXR-b), is a member of the nuclear receptor family of transcription factors. LXRs directly regulate the expression of many genes involved in fatty acid synthesis, including SREBP1, FAS, and several desaturases and fatty acid elongates required for the synthesis of long-chain polyunsaturated fatty acids [52]. The net effect of LXR activation is an increase in the levels of long-chain polyunsaturated fatty acids (PUFAs). Our finding that MALAT1 knockdown further reduces the expression of NR1H2 suggests that Ex-4-induced downregulation of MALAT1 leads to NR1H2 downregulation, which, in turn, decreases triglyceride and PUFA synthesis and reduces lipid accumulation.

A gene that exhibits pronounced susceptibility to MALAT1 silencing is CD36, also known as fatty acid translocase (FAT). Consistent with observations in prior investigations [53], the exposure of HepG2 cells to OA elicited a noteworthy elevation in CD36 expression compared to untreated cells.

Previous research has extensively elucidated the pivotal role of CD36 in facilitating fatty acid uptake [54,55]. Furthermore, additional investigations have demonstrated intriguing findings regarding the impact of CD36 overexpression in hepatic tissue, particularly in transgenic mouse models (CD36Tg) [56]. Contrary to expectations, studies have revealed that the overexpression of CD36 in the liver, as observed in CD36Tg mice, does not exacerbate metabolic dysregulation. Instead, these transgenic mice exhibit a remarkable resilience to the adverse metabolic alterations induced by high-fat diet (HFD) consumption and extended fasting periods. These noteworthy observations underscore the protective role of CD36 expression in hepatic metabolism. This intriguing phenomenon was elucidated in a study conducted by Wilson et al. [53], highlighting that CD36 holds functional significance beyond being merely indicative of disease; their results were obtained using mice with hepatocyte-specific deletion of Janus kinase 2 (JAK2L mice). Taken together, these findings illustrate the significance of CD36 as a critical mediator of hepatic fatty acid uptake, particularly in contexts characterized by elevated fatty acid levels such as OA, and the upregulation of CD36 following Ex-4 treatment offers protection against steatosis via the modulation of MALAT1 expression.

Although the full mechanism underlying the susceptibility of CD36 expression to MALAT1 silencing remains to be fully elucidated, it has been suggested that it is mediated through the miR-206/ARNT axis [33]. MiR206 is suggested to not only prevent both NAFLD and hyperglycemia progression [57] but also inhibit Srebp1c-induced lipogenesis and activate the insulin signaling pathway. On the other hand, ARNT, also known as HIF-1β, is suggested to negatively regulate the expression of PPARα, a key regulator that inhibits lipid accumulation in the liver, which can further cause inhibition of CD36 expression [58]. Remarkably, this upregulation is further potentiated in the presence of Ex-4, implying the involvement of CD36 in the observed Ex-4-treated steatosis reduction. Notably, while the silencing of MALAT1 does not significantly modulate the impact of OA treatment on CD36 expression, it effectively impedes the additional upregulation induced by Ex-4.

These findings prove that the Ex-4-mediated amplification of CD36 expression involves MALAT1. CD36 is a protein-coding gene that is a member of the class B scavenger receptor family that can bind long-chain fatty acids, phospholipids, collagen, and low-density lipoprotein [59]. It is involved in fatty acid uptake activity and has demonstrated an association with various metabolic diseases, such as atherosclerosis, diabetes mellitus, and obesity [60]. The localization of CD36 on the plasma membrane of hepatocytes was found to be markedly increased in patients with NASH compared to patients with non-steatotic livers and those with simple steatosis [60]. An increase in CD36 expression was also observed in HFD-induced NAFLD mice, and hepatocyte-specific disruption of CD36 attenuated fatty liver and improved insulin sensitivity in these mice [53]. Moreover, it has been suggested that CD36 can form a metabolic pathway through PPARγ and regulate lipid metabolism [61]. Recent investigations indicate a notable correlation between the increased expression of hepatic CD36 and insulin resistance, hyperinsulinemia, and heightened steatosis in NASH [62]. Furthermore, Zeng et al. [63] reported that palmitoylation of CD36 is significantly upregulated in NAFLD and that inhibition of this palmitoylation alleviates NAFLD by promoting CD36 localization to the mitochondria of hepatocytes. Their study suggested that CD36 functions as a molecular bridge between long-chain fatty acids and long-chain acyl-CoA synthetase 1 to increase the production of long-chain acyl-CoA, thus promoting fatty acid oxidation and inhibiting lipid accumulation in hepatocytes. More recently, it was suggested that CD36 is involved in SREBP1 processing and the lipogenic program in the liver by interacting with the insulin-induced gene 2 (INSIG2) and that hepatocyte CD36-mediated de novo lipogenesis is a critical factor in the development of NAFLD [63]. How MALAT1 regulates the expression of CD36 remains elusive. However, using VAX939, a chemical inhibitor of β-catenin, Huangfu et al. suggested that MALAT1 promotes CD36 transcription partly via β-catenin [64]. They also found that MALAT1 knockdown decreases β-catenin binding to the CD36 promoter and vice versa. Noteworthy, we [21] and others [22] previously showed that Ex-4-induced improvement of OA-induced steatosis in HepG2 implicates the Wnt/β-catenin pathway. A comprehensive understanding of the molecular landscape involving MALAT1, CD36, and lipogenic genes warrants further investigations and holds the potential to inform innovative therapeutic strategies aimed at mitigating the development and progression of NAFLD.

LncRNAs can interact with miRNAs through miRNA response elements (MREs), competing with miRNAs and reducing their availability to target genes. This interaction, known as the competitive endogenous RNA (ceRNA) network, plays a key role in regulating gene expression. MiRNAs are typically non-coding RNAs, about 22 nucleotides in length, that influence various biological processes. Notable miRNAs include miR-206 [57], miR-30b [65], and miR-130a [66], among others. MiR-206, in particular, has been shown to prevent the progression of both NAFLD and hyperglycemia [67]. Wu et al. demonstrated that miR-206 can inhibit Srebp1c-induced lipogenesis while promoting insulin signaling [57]. In this context, MALAT1 appears to influence hepatic lipid metabolism by sponging miR-206 to activate ARNT. Furthermore, downregulation of miR-206 can counteract the effects of MALAT1 knockdown on FFA-induced lipid accumulation in hepatocytes. However, despite these promising insights, the precise mechanisms by which MALAT1 regulates hepatic lipid metabolism remain unclear, warranting further investigation.

We recognize the limitations of using HepG2 cells as an in vitro model for studying steatosis, especially when exploring the mechanisms responsible for the protective effects of GLP-1RAs. As a result, the effects of Ex-4 observed at the cellular level may not fully capture its effects within the context of an entire organism. Although the role of MALAT1 and the impact of GLP-1RAs in steatosis have not yet been examined in vivo, there is substantial evidence supporting their ability to reduce the liver fat content in both animal models and patients with NAFLD.

## 5. Conclusions

In conclusion, the present study proposes that the improvement in OA-induced steatosis in response to Ex-4 involves the regulation of the expression of several genes, and that this regulation is mediated through MALAT1 (Figure 7). Modulating hepatic MALAT1’s expression or functionality could present a promising strategy for NAFLD management.

## Figures and Tables

**Figure 1 biomedicines-13-00370-f001:**
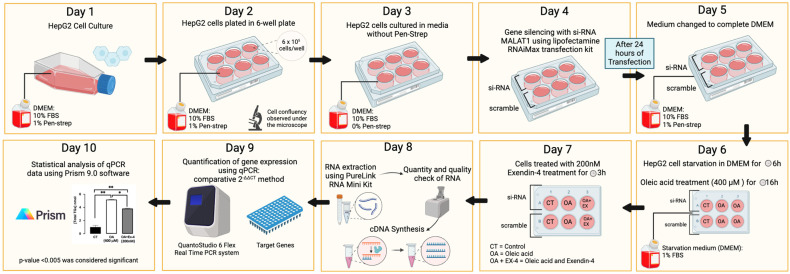
HepG2 cell culturing, MALAT1 transfection, and oleic acid and Exendin-4 treatment protocols. HepG2 cells were starved for 6 h and then treated with 400 µM of oleic acid for 16 h. Cells were then treated with 200 nM of Exendin-4 for 3 h. β-actin was used as a housekeeping gene during the qPCR. Significance levels are denoted by * *p* < 0.05, ** *p* < 0.01.

**Figure 2 biomedicines-13-00370-f002:**
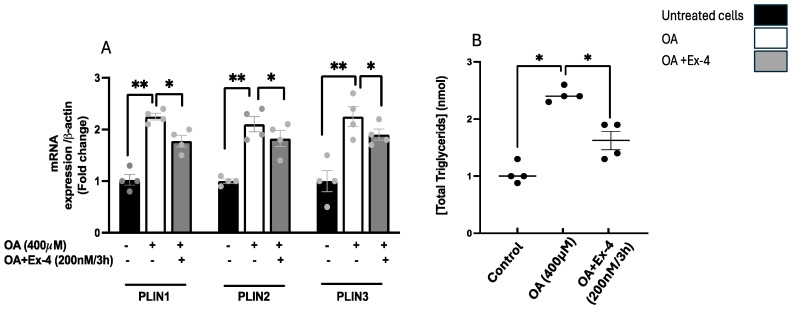
The impact of Exendin-4 on mitigating oleic acid-induced lipid accumulation in HepG2 cells. HepG2 cells were subjected to a 6 h starvation period, followed by treatment with oleic acid (OA; 400 μM) for 16 h, and subsequently treated for 3 h with OA with or without Exendin-4 (Ex-4; 200 nM). (**A**) Quantifications of mRNA expression levels of perilipin 1 (PLIN1), perilipin 2 (PLIN2), and perilipin 3 (PLIN3). The expression levels were normalized to the level of β-actin. (**B**) Exendin-4 significantly reduced the oleic acid-induced elevation in triglyceride (TG) contents within HepG2 cells. All data are presented as the mean ± standard error (SE) with *n* = 4. Significance levels are denoted by * *p* < 0.05, ** *p* < 0.01.

**Figure 3 biomedicines-13-00370-f003:**
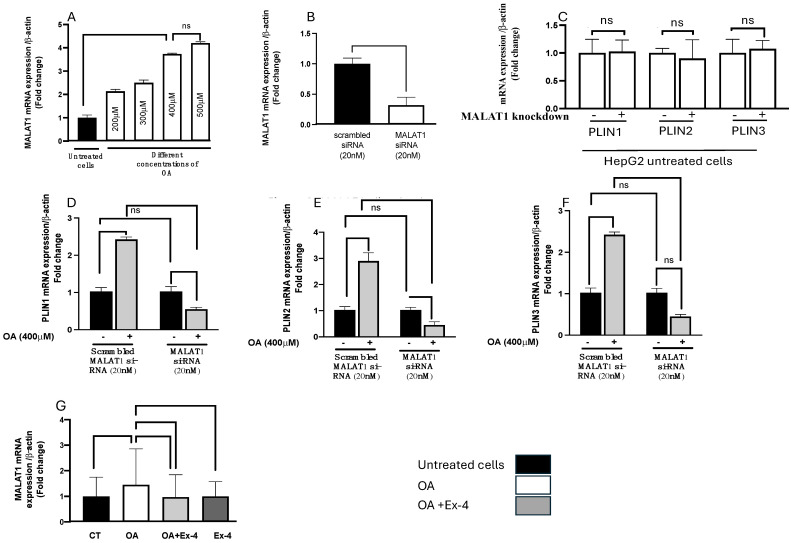
Silencing of MALAT1 counteracts the influence of oleic acid (OA) on HepG2 cells. (**A**) HepG2 cells were exposed to various concentrations of OA (200, 300, 400, 500 μM) for 24 h, after which the MALAT1 levels were measured, with white bars representing cells treated with OA and black bars representing control cells. (**B**) HepG2 cells were transfected with scrambled si-RNA or si-MALAT1 for 48 h, followed by the quantification of MALAT1 expression using qPCR. (**C**) MALAT1 knockdown did not alter the mRNA expression of the PLIN1, PLIN2, and PLIN3 genes in untreated HepG2 cells. (**D**–**F**) The mRNA expression of PLIN1, PLIN2, and PLIN3, respectively, in HepG2 cells transfected with scrambled si-RNA and si-MALAT1 for 24 h, starved for 6 h, and subsequently treated with oleic acid (OA; 400 μM) for 16 h. (**G**) MALAT1 expression in HepG2 cells with OA, OA+Ex-4, and Ex-4-alone treatment. The gene expressions of PLIN1, PLIN2, and PLIN3 were then quantified under MALAT-1 silencing. The results are depicted as the mean ± standard error (SE) based on six replicates. Statistical significance is denoted by ** *p* < 0.01, *** *p* < 0.001, **** *p* < 0.0001, and ns: not significant.

**Figure 4 biomedicines-13-00370-f004:**
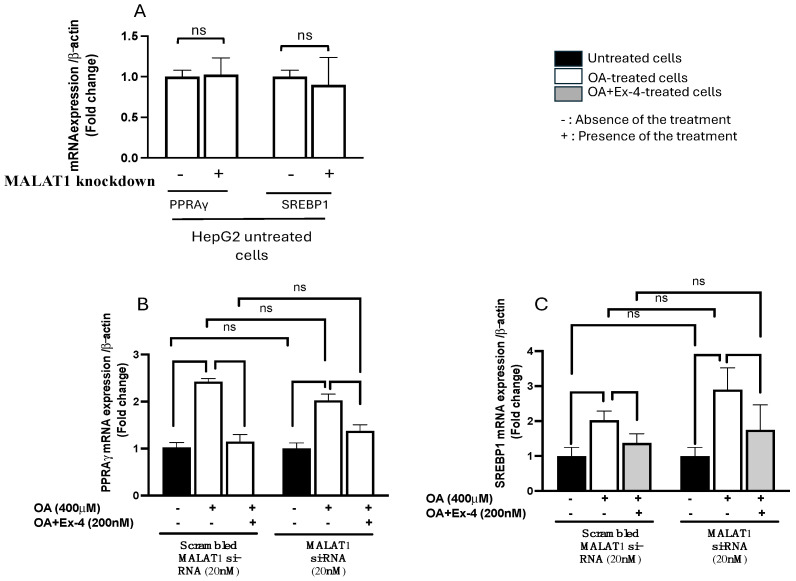
MALAT1 silencing does not affect de novo lipogenesis genes. The quantification of de novo lipogenesis gene expression was conducted using qRT-PCR, and the results were normalized to the level of β-actin. (**A**) MALAT1 transfection did not alter the mRNA expression of PPARγ and SREBP-1 genes in untreated HepG2 cells (control cells). (**B**,**C**) The expression levels of PPARγ (peroxisome proliferator-activated receptor gamma) and SREBP-1 (sterol regulatory element-binding transcription factor 1), respectively, transfected with either scrambled MALAT1 si-RNA or MALAT1 siRNA at a concentration of 20 nM. All data are expressed as the mean ± standard error (SE) with *n* = 6. Statistical significance is indicated by * *p* < 0.05, ** *p* < 0.01, and ns: not significant.

**Figure 5 biomedicines-13-00370-f005:**
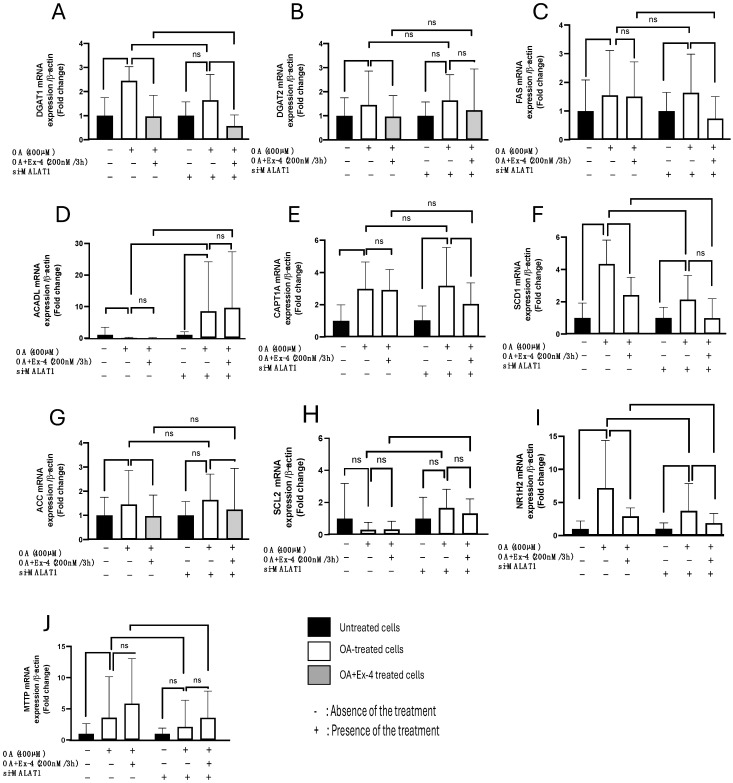
MALAT1 suppression alters the expression of lipogenesis-related genes in HepG2 cells. All genes were transfected with either scrambled MALAT1 si-RNA or MALAT1 siRNA for 24 h. Quantifications of the mRNA expression levels for DGAT1 (Diacylglycerol O-acyltransferase 1), DGAT2 (Diacylglycerol O-acyltransferase 2), FAS (fatty acid synthase), ACADL (Acyl-CoA Dehydrogenase Long Chain), CPT1A (Carnitine Palmitoyl Transferase 1A), SCD1 (Stearoyl-CoA desaturase 1), ACC (Acetyl-CoA carboxylase alpha), SCL2 (Collagen-like protein 2), FABP4 (fatty acid-binding protein 4), NR1H2 (nuclear receptor subfamily 1 group H member 2), and MTTP (Microsomal Triglyceride Transfer Protein) were performed using qPCR after treatment with OA alone or OA combined with Ex-4 (**A**–**J**). Data are presented as the mean ± standard error (SE) with *n* = 6. Statistical significance is indicated by * *p* < 0.05, ** *p* < 0.01, and ns: not significant.

**Figure 6 biomedicines-13-00370-f006:**
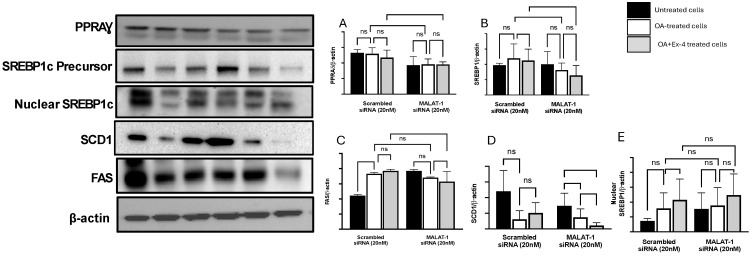
Western blotting and quantification of the following transcription factors: stearoyl-CoA desaturase 1 (SCD1), sterol regulatory element-binding protein-1 (SREBP1), the precursor and nuclear forms of SREBP1, peroxisome proliferator-activated receptor gamma (PPAR-γ), and FAS (fatty acid synthetase (**A**–**E**). Proteins were normalized against β-actin. All values are expressed as the mean SE. (*n* = 3). ns: not significant, * *p* < 0.05. Full-length blots are displayed in Data S1.

**Figure 7 biomedicines-13-00370-f007:**
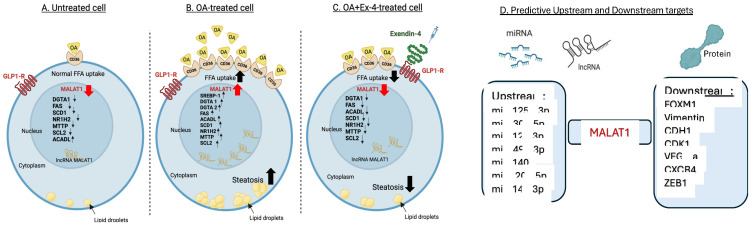
Proposed simplified signaling pathway elucidating the mechanism underlying the amelioration of steatosis induced by Exendin-4. At rest, when the levels of extracellular circulating FFAs are low, the uptake of these FFAs is low and the expression of MALAT1 and many lipid metabolism genes is downregulated. Following stimulation with OA, mimicking an increase in extracellular FFA levels, the expression of MALAT1 leads to the modulation of the expression of several genes, including SREBP-1, SCD1, FAS, ACC, DGAT1, DGTA2, MTTP, and NR1H2. In the presence of Ex-4 treatment, a downregulation of MALAT1 is associated with reversal of the modulations of numerous genes induced by OA, leading to a reduction in steatosis. Predictive upstream and downstream targets are highlighted. Upward and downward arrows indicate upregulation and downregulation, respectively.

**Table 1 biomedicines-13-00370-t001:** Primer list and sequences.

Gene	GenBank IDs	Forward Sequence (5′3′)	Reverse Sequence (5′3′)	PCR Product Sizes (pb)
*SREBP-1*	6720	GGCTCCTGCCTACAGCTTCT	CAGCCAGTGGATCACCACA	109
*PPARγ*	5468	GACCTCAGACAGATTGTCAC	AGTCCTTGTAGATCTCCTGC	106
*DGAT1*	8694	AACTGGTGTGTGGTGATGCT	CCTTCAGGAACAGAGAAACC	112
*DGAT2*	84649	CTACAGGTCATCTCAGTGCT	GAAGTAGAGCACAGCGATGA	120
*FAS*	355	TATGCTTCTTCGTGCAGCAGTT	GCTGCCACACGCTCCTCTAG	94
*ACADL*	33	TTGGCAAAACAGTTGCTCAC	ACATGTATCCCCAACCTCCA	123
*CPT1A*	1374	TCCAGTTGGCTTATCGTGGTG	CTAACGAGGGGTCGATCTTGG	244
*SCD-1*	20249	CACCACATTCTTCATTGATTGCA	ATGGCGGCCTTGGAGACT	75
*ACC*	104371	CAGAAGTGACAGACTACAGG	ATCCATGGCTTCCAGGAGTA	125
*NR1H2*	7376	GCGCTTGATCCTCGTGTAG	GCGCTTGATCCTCGTGTAG	639
*MTTP*	4547	CTACAGGTCATCTCAGTGCT	GAAGTAGAGCACAGCGATGA	120
*PLIN1*	5346	ACAGACCATTTCTCAGCTCCAT	TATCCAATGCTCCTTTTCCACT	141
*PLIN2*	123	GAACAGAGCTACTTCGTACG	CAGTTTCCATCAGGCTTAGG	151
*PLIN3*	10226	TGCTCCACTCACTTTACCGTC	TAGCGTCCAGTGTGTACTGAC	199
*MALAT1*	378938	GAATTGCGTCATTTAAAGCCTAGTT	GTTTCATCCTACCACTCCCAATTAAT	185
*β-actin*	60	TCATGAAGATCCTCACCGAG	CATCTCTTGCTCGAAGTCCA	116

## Data Availability

Data are contained within the article and Appendix A.

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
