# Peer review of "Exploring the Putative Involvement of MALAT1 in Mediating the Beneficial Effect of Exendin-4 on Oleic Acid-Induced Lipid Accumulation in HepG2 Cells"

_biomedicines, 2025, doi:10.3390/biomedicines13020370_

Round 1
Reviewer 1 Report
Comments and Suggestions for Authors
This study investigates the impact of MALAT1 on the efficacy of Exendin-4 in mitigating oleic acid-induced lipid accumulation in the HepG2 hepatocellular carcinoma cell line. The following comments and suggestions aim to improve the manuscript.
1. HepG2 cells, being a cancer cell line, exhibit distinct biological characteristics compared to immortalized hepatocytes. To more accurately model in vivo conditions, such as NAFLD, the use of immortalized hepatocyte cell lines like AML12 (mouse) or THLE-2 (human) would be more appropriate for mechanistic studies.
2. Ensure consistency in the units used for oleic acid (OA) concentration (mM or μM) throughout the manuscript.
3. To maintain consistency with the results described in lines 217-218, Figure 1 should visually represent the downregulation of PLIN2 and PLIN3.
4. Review the figures to ensure complete and accurate labeling of statistical analyses. Missing labels and x-axis lines in several panels require correction.
5. Include data on total triglyceride levels in MALAT1 knockdown (KD) cells in Figure 2.
6. In line 252, revise the figure reference to 'Figure 2C-2F'.
7. Include a description of the data presented in Figure 2G within the text.
8. Figure 5 should include data for both the precursor and nuclear forms of SREBP1c. Additionally, please verify the presence of the FAS Western blot in the supplemental data; it appears to be missing.
9. Revise Panel B of the graphical summary to accurately reflect the increased steatosis observed in OA-treated cells.
10. Investigate whether MALAT1 overexpression can abolish the beneficial effects of EX-4 on OA-induced steatosis.
Comments on the Quality of English LanguageThe English could be improved to more clearly express the research.
Author Response
Dr. Abdelilah Arredouani
Qatar Biomedical Research Institute
Hamad Bin Khalifa University
Doha, Qatar
aarredouani@hbku.edu.qa
January 27, 2025
Manuscript ID: biomedicines-3421462
Dear Dr. Roxana Valentina Stoian
We are pleased that our manuscript entitled ‘’Exploring the Putative Involvement of MALAT1 in Mediating the Beneficial Effect of Exendin-4 on Oleic Acid-Induced Lipid Accumulation in HepG2 Cells’’ (ID: biomedicines-3421462) has successfully passed the first review round and is now ready for another round of revision.
We would like to extend our sincere gratitude to the editorial board and the esteemed reviewers for their thorough evaluation of our manuscript. Their insightful comments and constructive critiques have been instrumental in improving the quality of our work.
After carefully reviewing each comment, we have carefully addressed all concerns raised by the reviewers and made the necessary revisions to the manuscript. Attached, you will find two versions of the revised manuscript: a clean copy and a highlighted version that shows all modifications made in response to the reviewers' feedback. We believe that these improvements will significantly enhance the quality and impact of our research findings, increasing the likelihood of favorable consideration for publication in your esteemed journal.
To facilitate the review process, we have included a detailed point-by-point response outlining our approach to addressing the reviewers' comments. This detailed response is provided below, with our responses clearly marked in red for easy reference.
We look forward to your valuable and positive feedback.
Yours sincerely,
On behalf of the co-authors
Dr. Abdelilah Arredouani

Reviewer 2 Report
Comments and Suggestions for Authors
The report
General Comments
The purpose of this manuscript is to examine how long non-coding RNA (LNCAR) MALAT1 interferes with the positive effects of Exendin-4, which is an agonist of glucagon-like peptide-1 receptor, on the buildup of lipid in HepG2 cells after being exposed to oleic acid. The study presents a vital topic for non-alcoholic fatty liver disease (NAFLD) and liver lipid metabolism. A detailed study framework is proposed by the authors, which includes qRT-PCR for gene expression analysis, siRNA-mediated MALAT1 silencing, and thallium quantification to evaluate lipid buildup. By data analyzing, we can understand the molecular mechanisms that underlie the therapeutic effects of GLP-1RAs in liver steatosis.
The study has a lot of possible for improvement in terms of rigor, clarity, and impact. I provide the manuscript's strengths and weaknesses below.
Major Comments
1. Scientific Rationale and Novelty
• Strengths: The study talks a novel and underexplored part by connecting MALAT1, a well-known lncRNA, to the effects of Ex-4 on liver steatosis. The stressed on MALAT1's role in lipid metabolism genes is mostly attention, as it engagements the gap between lncRNA biology and liver metabolic disease.
• Weaknesses: While the study originality is clear, the study does not satisfactorily elaborate on the broader impacts of the outcomes. For example, how might these results translate to in vivo models or clinical settings? Additionally, the introduction could better frame, the significance of MALAT1 in NAFLD farther its known roles in diabetes-related complications and cancer.
Recommendation: broaden the introduction to include a more inclusive discussion of MALAT1's potential roles in hepatic lipid metabolism and NAFLD. Highlight how this manuscript advances the area compared to previous study.
2. Experimental Design and Methodology
• Strengths: The experimental design is robust, with clear protocols for inducing steatosis, Ex-4 treatment, and MALAT1 silencing. The use of multiple assays (e.g., triglyceride quantification, qRT-PCR) strengthens the validity of the findings.
• Weaknesses:
o The manuscript does not provide sufficient detail on the controls used in the experiments. For example, were scrambled siRNA controls included to account for off-target effects of MALAT1 silencing?
o The duration of Ex-4 treatment (3 hours) seems relatively short. Was this time point chosen based on preliminary data? If so, this should be explicitly stated.
o The study focuses on HepG2 cells, which, while widely used, have limitations as a model for hepatic steatosis. The authors should acknowledge this limitation and discuss the need for validation in primary hepatocytes or animal models.
Recommendation: Provide additional details on experimental controls and justify the choice of treatment duration. Include a discussion of the limitations of the HepG2 cell model in the discussion section.
3. Data Presentation and Analysis
• Strengths: The data of the results are presented clearly, with appropriate use of tables and figures. The statistical analysis is complete, and the use of ANOVA with post hoc tests is adequate for the study design.
• Weaknesses:
o The manuscript does not include raw data or representative images for key experiments, such as lipid droplet staining with BODIPY. Including such images would strengthen the visual presentation of the results.
o The proposed signaling pathway in Figure 5 is interesting but oversimplified. For example, it does not account for potential upstream regulators of MALAT1 or downstream effects beyond lipid metabolism.
Recommendation: Include representative images of lipid droplet staining and provide more detailed annotations in Figure 5 to reflect the complexity of the signaling pathway.
4. Interpretation of Results
• Strengths: The authors provide a thoughtful interpretation of the data, particularly regarding the differential effects of MALAT1 silencing on lipid metabolism genes.
• Weaknesses:
o The discussion part does not sufficiently address why MALAT1 hushing did not affect de novo lipogenesis genes (e.g., SREBP1, PPARγ) but inclined other lipid metabolism genes (e.g., ACADL, FAS, CPT1A). This difference permits further exploration.
o The manuscript does not discuss potential off-target effects of Ex-4 or MALAT1 silencing, which could confound the results.
Recommendation: Provide a more detailed discussion of the differential effects of MALAT1 silencing on lipid metabolism genes. Address the possibility of off-target effects and suggest future experiments to validate the findings.
Minor Comments
1. Typographical Errors:
o Line 17: "sudy" should be changed to "study."
o Line 20: "was was" should be changed to "was."
o Line 24: "such FAS" should be changed to "such as FAS."
o Line 25: "invetigations" should be changed to "investigations."
2. Abbreviations:
o Please, sure all abbreviations are defined in the first use (e.g., triglycerides for TGs, free fatty acids for FFA).
3. References:
o Some references are outdated or incomplete. For example, reference 60 (Alkhatatbeh et al.) is from 2016, and more recent studies on OA-induced steatosis models could be cited.
4. Ethical Statement:
o The manuscript mentions ethical approval for animal or human studies but does not clarify whether this study involved such experiments. If not applicable, this should be explicitly stated.
Summary of Recommendations
1. Expand the introduction to better contextualize the significance of MALAT1 in NAFLD.
2. Provide additional details on experimental controls and justify the choice of treatment duration.
3. Include representative images of lipid droplet staining and improve the signaling pathway figure.
4. Discuss the differential effects of MALAT1 silencing on lipid metabolism genes and address potential off-target effects.
5. Include a data availability statement and supplementary materials.
6. Correct typographical errors and ensure consistent use of abbreviations.
Author Response

(The authors gave the same response as above.)

Round 2
Reviewer 1 Report
Comments and Suggestions for Authors
The revised manuscript comprehensively addresses the principal concerns raised in the previous review. Substantial improvements in clarity, conciseness, and overall quality that make it suitable for publication in Biomedicines.